# TcdB of *Clostridioides difficile* Mediates RAS-Dependent Necrosis in Epithelial Cells

**DOI:** 10.3390/ijms23084258

**Published:** 2022-04-12

**Authors:** Florian Stieglitz, Ralf Gerhard, Rabea Hönig, Klaudia Giehl, Andreas Pich

**Affiliations:** 1Institute of Toxicology, Hannover Medical School, Carl-Neuberg-Str. 1, 30625 Hannover, Germany; stieglitz.florian@mh-hannover.de (F.S.); gerhard.ralf@mh-hannover.de (R.G.); 2Core Facility Proteomics, Hannover Medical School, Carl-Neuberg-Str. 1, 30625 Hannover, Germany; 3Signal Transduction of Cellular Motility, Internal Medicine V, Justus Liebig University Giessen, Aulweg 128, 35392 Giessen, Germany; rabea.hoenig@innere.med.uni-giessen.de (R.H.); klaudia.giehl@innere.med.uni-giessen.de (K.G.)

**Keywords:** *Clostridioides difficile*, *C. diff.*, TcdB, phosphoproteomics, RAS, TcdA, pyknosis, glucosyltransferase-independent effect

## Abstract

A *Clostridioides difficile* infection (CDI) is the most common nosocomial infection worldwide. The main virulence factors of pathogenic *C. difficile* are TcdA and TcdB, which inhibit small Rho-GTPases. The inhibition of small Rho-GTPases leads to the so-called cytopathic effect, a reorganization of the actin cytoskeleton, an impairment of the colon epithelium barrier function and inflammation. Additionally, TcdB induces a necrotic cell death termed pyknosis in vitro independently from its glucosyltransferases, which are characterized by chromatin condensation and ROS production. To understand the underlying mechanism of this pyknotic effect, we conducted a large-scale phosphoproteomic study. We included the analysis of alterations in the phosphoproteome after treatment with TcdA, which was investigated for the first time. TcdA exhibited no glucosyltransferase-independent necrotic effect and was, thus, a good control to elucidate the underlying mechanism of the glucosyltransferase-independent effect of TcdB. We found RAS to be a central upstream regulator of the glucosyltransferase-independent effect of TcdB. The inhibition of RAS led to a 68% reduction in necrosis. Further analysis revealed apolipoprotein C-III (APOC3) as a possible crucial factor of CDI-induced inflammation in vivo.

## 1. Introduction

A *Clostridioides difficile* infection (CDI) is the most common nosocomial infection worldwide [1]. CDI symptoms extend from asymptomatic cases to diverse forms of diarrhea. In severe cases, a CDI could lead to pseudomembranous colitis with a fatal outcome. The Centers for Disease Control and Prevention listed *Clostridioides difficile* among the US’s top five urgent threats in 2019 [2].

The main virulence factors are TcdA and TcdB, which belong to the family of the large clostridial glucosylating toxins [3]. TcdA and TcdB consist of four domains, the receptor-binding domain (RBD), a translocation domain (TD), an autoprocessing domain (AD) and a glucosyltransferase domain (GTD) [3]. For TcdA sulfated glycosaminoglycans (sGAGs), gp96, low-density lipoprotein receptor (LDLR) and low-density lipoprotein related protein 1 (LRP1) have been described as receptors [4,5,6]. Chondroitin sulfate proteoglycan 4 (CSPG4), Nectin 3 and frizzled protein (FZD1, 2 and 7) have been described as the main receptors for TcdB [7,8,9]. After endocytosis, the endosomal acidification leads to the activation of the TD. Subsequently, the subunits AD and GTD are translocated through an endosomal pore into the cytosol. The AD domain cleaves the GTD subunit off the holotoxin and releases the GTD into the cytosol [10]. Rho-GTPases are the primary substrate for the GTD. The glucosylation of Rho-GTPases leads to a reorganization of the actin cytoskeleton, cell rounding and, subsequently, apoptosis, leading to a breakdown of the colon epithelial barrier [3]. The process of glucosylation and apoptosis is well understood and has been extensively studied by our group and others [11,12,13,14,15].

Interestingly, TcdB also exhibits rapid necrotic cell death at toxin concentrations above 0.1 nM in vitro, which is called pyknosis [16,17]. This effect has been described as glucosyltransferase independent since the glucosyltransferase-deficient variant TcdB_NXN_ also induces this pyknotic phenotype. This effect is also independent of the AD subunit [16,17]. Pyknosis is characterized by chromatin condensation, nuclear dissolution and ROS production [16]. Pyknosis is visually characterized by cell shrinkage, chromatin condensation and membrane blebbing [16]. Although the glucosyltransferase-dependent effects are the leading cause for CDI-associated symptoms in vivo [18], the necrotic cell response might act synergistically to promote inflammation and contribute to the aftermath of a CDI, for example, irritable bowel syndrome [19,20]. Several studies have indicated that NADPH complex assembly and subsequent ROS production lead to pyknosis, but the underlying signaling cascade is unclear [21,22].

This study aimed to elucidate the underlying mechanism of the glucosyltransferase-independent effect of TcdB. We conducted a large-scale analysis of the phosphoproteome of the HEp-2 cell line treated with TcdB, TcdB_NXN_, and for the first time, TcdA and TcdA_NXN_ in a multiplex tandem mass tags (TMT)-based approach with the overall goal of finding regulators of the glucosyltransferase-independent effect.

## 2. Results

### 2.1. Morphological Changes after Treatment with Toxins

For the evaluation of the effects of toxins on the phosphoproteome of HEp-2, cell concentrations of 20 nM for TcdA and TcdA_NXN_ and 2 nM for TcdB and TcdB_NXN_ were used to be consistent with prior analysis of the proteome of HEp-2 cells after treatment with TcdA, TcdA_NXN_, TcdB and TcdB_NXN_ from our group [13,14,22]. After 8 h of toxin treatment, TcdA- and TcdB-treated cells showed a round morphology, and TcdB- and TcdB_NXN_-treated cells also displayed pyknotic cell death, while untreated controls and TcdA_NXN_-treated cells remained unchanged in their morphology (Figure 1A). Additionally, pyknotic cell death was analyzed through DAPI staining for TcdB and TcdB_NXN_. TcdB- and TcdB_NXN_-treated cells that exhibited a pyknotic phenotype were also DAPI positive (Appendix A).

### 2.2. Comparative Phosphoproteomic Analysis

For the first analysis, to investigate the glucosyltransferase-independent effect, we used a single 8 h time point TMT 6-plex-based approach. Cells were treated with TcdA, TcdA_NXN_, TcdB and TcdB_NXN_. Untreated cells served as the control. We could detect 14,242 phosphosites across all three replicates with a localization probability >75% and 4530 protein groups. For further analysis, we included only phosphosites that could be normalized on the corresponding protein. After normalization, 11,282 phosphosites were included in further analysis. To determine whether the data were suitable for further analysis, a principal component analysis (PCA) was performed, which reduced the data to two components that explained most of the changes that occurred in the dataset (Figure 1B). Our analysis dissected all different treatments (TcdA, TcdA_NXN_, TcdB, TcdB_NXN_ and untreated control) from one another in the PCA, although TcdA_NXN_-treated cells and the untreated cells clustered very closely together and were, therefore, interpreted as unchanged.

As an initial threshold, we considered only those phosphosites that were significantly (*p* < 0.05) altered and showed an at least 2-fold regulation compared to the control as changed. We found the following numbers of phosphosites that fulfilled these criteria: 1086 phosphosites for TcdB-treated cells, 936 for TcdB_NXN_-treated cells, 414 for TcdA-treated cells and 353 for TcdA_NXN_-treated cells. The 20 most significantly regulated phosphosites are listed in Appendix A. Subsequently, we analyzed all significantly (*p* < 0.05) 2-fold regulated phosphosites from all conditions with Ingenuity Pathway Analysis (IPA) software. The main downregulated canonical pathways for TcdA and TcdB were “Signaling of Rho Family GTPases” and “Actin Cytoskeleton Signaling”. Interestingly, the “14-3-3-mediated Signaling” and “HER-2 Signaling in Breast cancer” pathways were both highly upregulated in TcdB-treated cells, while there was no regulation found for those pathways in TcdA-treated cells. Similar to TcdB treatment, TcdB_NXN_-treated cells also exhibited a strong upregulation in “HER-2 Signaling in Breast cancer” along with a strong upregulation in “EGF Signaling”. TcdA_NXN_-treated cells only showed a few significantly enriched pathways (Appendix A).

Cells treated with TcdA exhibited no glucosyltransferase-independent effect and, therefore, served as a control for the glucosyltransferase-independent effects of TcdB. By subtracting the TcdA effects on the phosphoproteome from those of TcdB-treated cells the underlying glycosyltransferase-independent effect could be determined (Figure 1C). Phosphosites (4830) that were significantly (*p* < 0.05) altered between TcdA and TcdB were taken for further analysis; those phosphosites were then filtered for their two-fold upregulation between TcdB_NXN_ and control cells, resulting in 2375 phosphosites that were then used to perform a network analysis with the string db database. MAPK1 and EGFR emerged as central regulated kinases, both showing significantly upregulated phosphosites after TcdB and TcdB_NXN_ treatment, respectively (Figure 1D,E). Additionally, the phosphorylation of proteins that regulate the chromatin structure, mRNA processing, translational initiation and actin binding was altered (Figure 1F).

### 2.3. Kinetics of Phosphoproteome Alteration Point to RAS as Central Upstream Regulator

Next, we performed a kinetic experiment with different time points for TcdB_NXN_ and TcdB to evaluate glucosyltransferase-independent effect in more detail. We could detect 7595 phosphosites that were present in at least three out of four replicates and 5033 protein groups. After normalization on the corresponding protein, 6668 phosphosites remained for further analysis. We analyzed the data via PCA and found the least changes at the 20 min time point where some clustering of TcdB and TcdB_NXN_ versus the controls were obvious but no clear separation between TcdB and TcdB_NXN_ treatment could be achieved (Figure 2). After 1 h of treatment, all groups could be clearly separated, while TcdB_NXN_ still clustered closer to the control than to TcdB. The phosphoproteome of TcdB- and TcdB_NXN_-treated cells clustered closer together at the 2 h, 3 h and 8 h time points, although for the 3 h time point, no clear separation between TcdB and TcdB_NXN_ could be achieved. These separation patterns correspond to the significant 2-fold changes in the phosphoproteome of with TcdB_NXN_- and TcdB-treated HEp-2 cells. Only small changes compared to controls in the phosphoproteome of TcdB_NXN_-treated cells could be observed after 20 min and 1 h, but for TcdB-treated cells, major changes in the phosphoproteome took place as early as 1 h (Appendix A).

Subsequently, an IPA upstream comparison analysis of significantly changed phosphosites for TcdB and TcdB_NXN_ treatment was performed to identify a possible signaling cascade for the glucosyltransferase-independent effect (Figure 3A,B). IPA predicts potential upstream regulators based on upregulated or downregulated phosphorylation sites and is able to compare these upstream regulators between different datasets. Most downregulated upstream regulators, CDK2, CDK6, SRPK2 and P-TEFb, were predicted for both treatments. Activated upstream regulators DGKZ, IKBKE and RAC1 were identified for both conditions, whereas RAF1 was predominantly activated under TcdB_NXN_ treatment. To find a connection between the upstream regulators RAC1 and RAF1, the downstream targets of both were analyzed (Figure 3C). For RAF1, the MAPK1 (T185) phosphorylation was analyzed and showed the same activation pattern for TcdB- and TcdB_NXN_-treated cells. For RAC1, the phosphorylation pattern of PAK2 (S141) was analyzed. PAK2 phosphorylation in TcdB-treated cells was transiently upregulated, being significantly 4-fold upregulated after 1 h. TcdB_NXN_ treatment caused a constant upregulation in the PAK2 phosphorylation after 2 h. Interestingly, the phosphorylation pattern of AKT1S1 (S183) mimicked the pattern of PAK2 in such a way that TcdB treatment led to an initial strong upregulation at 1 h and shrank afterwards. However, in contrast to PAK2, the phosphorylation remained at a higher significant level. The phosphorylation pattern of PIK3C3 (S244) remained similar in both treatments and was not affected by PAK2 activation (data not shown). Therefore, the observations led us to the hypothesis for RAS as a connecting upstream regulator of the observed downstream phosphorylation patterns (Figure 3A).

### 2.4. RAS Inhibition Led to a Reduction in Pyknosis

HEp-2 cells were preincubated for 17 h with 1 µM of the pan-RAS inhibitor 3144 (3144) and afterwards treated for 8 h with TcdB or TcdB_NXN_ to assess the impact of RAS inhibition on the glucosyltransferase-independent effect (Figure 4A). Cells were then subjected to flow cytometry analysis. The cell pool pretreated with the inhibitor showed a significant 61.9% mean decrease in propidium iodide (PI)/annexin-positive cells when incubated with TcdB, and a 68% reduction was evident when treated with TcdB_NXN_ (Figure 4B, Appendix A). Preincubation with 20 nM TcdA led to a pronounced and significant increase in PI/annexin-positive cells for TcdB and TcdB_NXN_ treatment, but also in a small and significant increase in the control groups (Figure 4C). Pretreatment with 0.1 pM TcsL led to a reduction in PI/annexin-positive cells of 31.1% for TcdB_NXN_ and 51.9% for TcdB (Figure 4D). TcsL and TcdA were used as control for RAS inhibition. TcsL glucosylates its main substrate RAS, whereas TcdA inactivates RAS only after longer incubation times and preferentially modifies other small Rho-GTPases. For a more detailed explanation, please see the Section 3 of this paper. The observed findings also corresponded to the microscopic observations demonstrating a reduced occurrence of the pyknotic phenotype (Figure 4A). The DAPI staining of cells pretreated with the pan-RAS inhibitor 3144 also corresponded to the decrease in pyknotic cells observed using flow cytometry (Appendix A).

### 2.5. RAS Is Activated after TcdB Treatment

To evaluate the effects of TcdB and TcdB_NXN_ RAS activity RAS*GTP pull-down assays were performed. The phosphorylation of MAPK1/3 at (Thr202/Tyr204) was analyzed in the same lysate using a Western blotting procedure. MAPK1/3 is a downstream effector of activated RAS; therefore, its activation status was monitored as an additional marker for RAS activation (Figure 5B). Significant RAS activation after TcdB treatment could be confirmed after 3 min and 10 min of treatment and declined gradually within 60 min. For TcdB_NXN_, we found no elevated RAS*GTP levels for the investigated time points (Figure 5A). For TcdB treatment, a strong but non-significant upregulation of MAPK1/3 phosphorylation was evident at 30 min and further increased at 60 min (Figure 5B). For TcdB_NXN_, enhanced MAPK1/3 phosphorylation was observed at 60 min.

### 2.6. RAS Inhibition Influences Differences in Motif Phosphorylation and Reduced APOC3 Expression

We conducted an 8 h time point 6-plex phosphoproteome experiment to investigate the impact of RAS inhibition on the effect of TcdB and TcdB_NXN_ treatment. We could detect 3336 phosphosites with a localization probability of >75% for 1118 protein groups. All significantly altered phosphosites were included in our analysis to obtain a broad overview of phosphorylation changes in this experiment. Analysis via the supervised clustering of Benjamini Hochberg FDR-based ANOVA significantly regulated phosphosites revealed that all triplicates of each condition clustered together, except for the controls. The controls clustered in a larger column assigned as cluster 4 (Figure 6A). The main differences between the phosphorylation status of the DMSO and 3144 pretreated HEp-2 cells that were incubated with TcdB or TcdB_NXN_ resembles row cluster 6, which comprised a special set of proteins (Figure 6A). Enrichment analysis of this row cluster via Fisher’s exact test showed that “RNA processing” was the most significantly enriched gene ontology (GO)-term. Further enriched motifs such as “Casein kinase I substrate motif”, “PKC kinase substrate motif”, “PKA kinase substrate motif” and “MAPKAPK2 kinase substrate motif” were also significantly enriched motifs (Appendix A). To further examine the impact of the RAS inhibition on the glucosyltransferases effect, we conducted a Fisher exact test on all ANOVA significant phosphosites and filtered motifs that had the same direction in their mean phosphorylation status between TcdB and TcdB_NXN_. These should resemble the underlying glucosyltransferase-independent effect and be comparable to the so-called “co-phosphorylated” motifs between the DMSO and 3144 pretreated cells. The motifs “PKC kinase substrate motif”, “PKA kinase substrate motif”, “MAPKAPK2 kinase substrate motif”, “MDC1 BRCT domain binding motif”, “Plk1 PBD domain binding motif” and “GSK3 kinase substrate motif” showed the strongest decrease in their mean phosphorylation when the cells were pretreated with the pan-RAS inhibitor 3144, indicating that they should be involved in glycosyltransferase-independent effects. The other substrate motifs also showed a slight decrease in their mean phosphorylation (Figure 6B).

The phosphoproteome of HEp-2 cells treated with 2 nM TcdB or TcdB_NXN_ for 8 h showed broad changes, but the proteome itself was marginally changed. Noteworthy, only one protein, namely, apolipoprotein C3 (APOC3), showed a 2-fold reaction upon TcdB or TcdB_NXN_ treatment. The APOC3 expression increased linearly over time and was significantly >2-fold upregulated upon TcdB or TcdB_NXN_ treatment after 8 h (Figure 7A). APOC3 protein is involved in triglyceride-rich lipoprotein metabolism and has recently been shown to induce sterile inflammation [23]. Interestingly, this upregulation could be significantly inhibited by preincubation with the RAS inhibitor 3144 (Figure 7B).

## 3. Discussion

This study conducted a large-scale phosphoproteomic analysis on the glucosyltransferase-independent effect of TcdB on HEp-2 cells. In prior studies, TcdA and the glucosyltransferase-deficient variant TcdA_NXN_ showed no glucosyltransferase-independent necrotic effect [15], which is also in coherence with our findings (Figure 1A,B). Therefore, we subtracted the impact of TcdA on the proteome and phosphosproteome from the effect of TcdB to extract the glucosyltransferase-independent effect. Thus, TcdA was used as the control. Afterwards, we verified the significant TcdB-induced changes in the phosphoproteome by once again filtering the 2-fold significantly upregulated phosphosites of TcdB_NXN_ compared to the TcdA control. Identified and altered phosphosites should include only those specific to the glucosyltransferase-independent effects of TcdB, which were verified by different calculations from the two experiments. Significantly regulated phosphosites were used for a network analysis and depicted MAPK1 and EGFR as the central regulating kinases involved in chromatin remodeling and translational processes (Figure 1D,E). While T185 is an activation site of MAPK1, T693 (T669) is an inhibiting phosphosite of EGFR phosphorylated by MAPK1 as part of a negative feedback loop following initial activation [24,25]. Both sites (T185 and T693 (T669)) were more strongly phosphorylated as a response to TcdB and TcdB_NXN_ treatment (Figure 1E and Figure 3C).

Furthermore, ingenuity pathway analysis of TcdB- and TcdB_NXN_-responsive phosphosites revealed the strong activation of the “HER-2 Signaling in Breast cancer” pathway, confirming the involvement of the EGFR-MAPK pathway in the glucosyltransferase-independent effect. These findings correspond to other studies, pointing out the involvement of MAPK-kinase in the glucosyltransferase-independent effect [26]. Furthermore, TcdA induced no activation of MAPK1 and EGFR or related pathways, verifying the role of MAPK1 and EGFR in the glucosyltransferase-independent effects. It is also worth mentioning that pathways “Signaling by Rho Family GTPases” and “Actin Cytoskeleton Signaling” were downregulated for TcdB and TcdA treatment. In contrast, TcdB_NXN_- and TcdA_NXN_-treated cells showed no downregulation of these signaling hubs since Rho-GTPases were not inactivated by these mutant toxins. This observation underlines the consistency of the obtained and calculated datasets.

The kinetic experiments showed that significant changes in the phosphoproteome occurred already 1 h after TcdB addition and 2 h after TcdB_NXN_ treatment (Figure 2 and Appendix A). These changes correlated with morphological observations that the occurrence of the pyknotic phenotype for TcdB_NXN_ is delayed by 1 h as compared to TcdB. This fact is also in coherence with prior studies [22]. A possible explanation for this could be a lower activity of the TcdB_NXN_ mutant or a slower uptake due to a lesser receptor affinity even at higher concentrations. Thus, the pyknotic phenotype occurred between 1 h and 2 h (data not shown).

The analysis of the top 10 upstream regulators with IPA also showed coherence with the literature. CDK6, CDK2, P-TEFb and SRPK2 were predicted to be inhibited in their activation status (Figure 3B). These kinases are mainly involved in cell cycle progression [27,28,29], which has been shown to be inhibited through TcdB [30]. RAC1 and IKBKE were predicted to be upregulated in their activation status (Figure 3B). RAC1 has been previously described as a potential upstream regulator of the glucosyltransferase-independent effect [22], IKBKE is a key regulator of NFκB activation [31] and NFκB has been recently described to be activated in a TcdB setting [32].

Interestingly, RAF1 was strongly activated after TcdB_NXN_ treatment but only slightly after TcdB treatment. Additionally, the involvement of RAF1 in TcdB-induced signaling has not been described so far. The phosphorylation pattern of several down- and upstream targets were analyzed to investigate a possible connecting regulator between RAC1, RAF1 and IKBKE. MAKP1 is a classical downstream target of RAF1 [33], and both phosphorylation patterns were very similar between TcdB and TcdB_NXN_ treatment. PAK2 was chosen to analyze the impact of RAC1 activation. The patterns of both treatments differed from each other. PAK2 was initially activated by RAC1 after 1 h of TcdB treatment but then the phosphorylation status decreases to the starting point, because RAC1 was inactivated by TcdB after 1 h [13]. PAK2 activation increased after TcdB_NXN_ treatment within 1 h, but phosphorylation did not decrease because TcdB_NXN_ cannot inhibit RAC1.

AKT1S1 showed a mixture of MAPK1 and PAK2 activation patterns, with a strong but transient initial activation by TcdB followed by a weaker but still persistent significant activation with a more than 2-fold increase in phosphorylation. The strong initial activation is due to PAK2 activation, potentially activating mTOR [34], which has been described to influence S183 phosphorylation on AKT1S1 [35]. The activation of AKT1S1 could only be explained through an upstream regulator that acts independently of RAC1, which is PIK3C3 [36]. PIK3C3 showed the same activation pattern as MAPK1. Connecting RAC1, RAF1 and PIK3 led to the hypothesis that RAS should be a potential upstream regulator of the glucosyltransferase-independent effect.

RAS inhibition led to a substantial and significant decrease in the pyknotic phenotype compared to the DMSO control (Figure 4B and Appendix A). Moreover, preincubation with TcdA, which does not glucosylate RAS as fast as other small Rho-GTPases [13], and with TcsL, whose main target is RAS [37], enabled us to exclude RAC1 as a central upstream regulator, as well as other small GTPases that could potentially be off-target effects of the pan-Ras-inhibitor 3144, thus leading us to false conclusions in the present study. Since RAC1 has been proven as a regulator in selected studies [21,22], we suggest RAS as an upstream regulator of RAC1 or a well-regulated cross talk of both GTPases as the central upstream regulator of the pyknotic effect.

TcdA increased the pyknotic as well as apoptotic cell death in TcdB- and TcdB_NXN_-treated cells (Figure 4C and Appendix A). TcdA also induced early-stage apoptotic cell death, which was further enhanced through additional treatment with TcdB and TcdB_NXN_ (Figure 4C and Appendix A). Apoptotic cell death is described for TcdA to occur after 24–48 h; therefore, findings are in line with prior studies [38]. Since oxidative stress accelerates apoptotic induction [39], enhanced early-stage apoptosis after TcdB or TcdB_NXN_ treatment due to additional ROS production in TcdA pretreated cells could be expected, leading to a higher PI- or annexin-positive rate in addition to the pyknotic effect (Appendix A).

While TcdA seemed to enhance pyknotic cell death, TcsL reduced it (Figure 4D). A possible explanation for this could be that the downregulation of RAC1, as conducted in previous studies via siRNA or knockdowns of RAC1 protein expression [20,40], impairs RAS signaling so that the pyknotic phenotype does not occur. A recent study shows that TcdB from the reference strain VPI10463 has binding capabilities to RAS but does not catalyze the glucosylation of R-RAS [41]. Additionally, TcdB from variant *C. difficile* strain 1470 serotype F (TcdBF) does not induce a pyknotic phenotype. TcdBF differs in its substrate specificity to the VPI10463 reference TcdB as it glucosylates R-RAS, in addition to RAC1 [42]. These findings lead to the hypothesis that TcdB might form complexes with RAS and RAC1 to promote the enhanced activation of RAS signaling. This was not analyzed in the present study but will be addressed in the future.

Furthermore, we showed that RAS was significantly activated 3 and 10 min after TcdB treatment, and this finding strengthens our suggestion for RAS as a potential central upstream regulator. However, we found no regulation of RAS activation after TcdB_NXN_ treatment at the investigated time points, although pan-RAS inhibitor reduced TcdB_NXN_ effects (MAPK, Calmodulin and 14-3-3 (Figure 6B), thus implying RAS dependency. A possible explanation for this could be that RAS activation occurred between 30 min and 1 h after TcdB_NXN_ treatment, which correlates with the strong MAPK1/3 activation after 1 h. This clearly indicates a later RAS activation by TcdB_NXN_ compared to TcdB for which RAS activation was immediately followed by MAPK1/3 phosphorylation (Figure 5B).

Interestingly, later time points showed no RAS activation (data not shown). A possible explanation for this could be that RAS activation steers the signaling towards the pyknotic pathway within the first hour and thus activates downstream signals as calcium signaling and strong ROS production that then drives the pyknotic effect with no further RAS activation. Since we did not control for toxin translocation, we could not clearly determine whether the RAS activation could have also been due to receptor binding. Nevertheless, TcdB_NXN_ was less potent than the wild-type TcdB in causing the pyknotic effect (Figure 4B). The RAS activation for TcdB was only narrowly visible, and the RAS activation of TcdB_NXN_ could have been below the detection line. If the RAS activation was receptor mediated, TcdB and TcdB_NXN_ should have produced the same activation signal since the mutations of TcdB_NXN_ are in the glucosyltransferase domain and not in the binding region of TcdB.

To examine the impairment effect of the pan-RAS inhibitor 3144 (3144) on the glucosyltransferase-independent effect, we conducted an 8 h-time point phosphoproteomic experiment. Activation patterns of kinases can be assessed by analyzing the phosphorylation of their specific motifs. Motifs are prevalent biochemical patterns around phosphorylation sites [43]. The global analysis of co-phosphorylated motifs of 3144 and toxin-treated cells showed that MAPKAPK2 motifs were strongly decreased in their mean phosphorylation compared to the TcdB and TcdB_NXN_ DMSO samples (Figure 6B) due to the central role of RAS in MAP kinase pathways [44]. This effect was reasonable if RAS was inhibited. Additionally, the “GSK3 kinase substrate motif” was downregulated. GSK3 plays an important role in the regulation of the NFκB pathway and is activated by the MAP kinase pathway [45]. Thus, if MAP kinase pathways are downregulated, the downregulation of GSK3 is consistent. Nevertheless, a possible RAS-independent activation of MAP kinase pathways could have also occurred since we found no RAS activation following TcdB_NXN_ treatment, despite strong MAPK1/3 signaling. Supporting an additional RAS-independent mechanism for pyknotic cell death is the finding that RAS inhibition through pan-RAS inhibitor 3144 only leads to a reduction in and not the elimination of the pyknotic effect.

PKC and PKA substrate motifs were annotated with the same motifs using Perseus software, so the mean regulation was the same for both proteins. The same was true for the “MDC1 BRCT domain binding motif” and for “Plk1 PBD domain binding motif”.

Interestingly, PKC is described in a recent study as an upstream regulator of the glucosyltransferase-independent effect [46]. We also found the phosphorylation of the PKC substrate motif to be enhanced by treatment with TcdB and TcdB_NXN_ and inhibited when cells were pretreated with 3144 (Figure 7B). Studies show that PKC could influence RAS signaling pathways [40], and RAS influences PKC activation through PIK3C [47]. Our studies showed that RAS inhibition with the subsequent inhibition of PIK3C led to a downregulation of PKC activity and the mean phosphorylation of PKC motifs (Figure 6B). MDC1 and BRCT are mainly involved in DNA damage response [48]. TcdB has been described to induce DNA damage through the glucosyltransferase-independent effect [30]. Inhibiting this glucosyltransferase-independent effect led to a reduced activation of MDC and BRCT, as shown in Figure 6B.

Analysis via the supervised hierarchical clustering of Benjamini Hochberg FDR-based ANOVA significant phosphosites showed that a distinct row cluster of proteins mainly separated 3144 pretreated cells after TcdB and TcdB_NXN_ treatment (Figure 6; row cluster 6). TcdB- and TcdB_NXN_-treated cells showed a pronounced upregulation in row cluster 6, whereas only 3144 pretreated cells showed no difference to the controls. Based on the motif enrichment analysis of the protein level, row cluster 6 (Figure 6A) mainly involved kinases that were already enriched in the global enrichment analysis (Appendix A and Figure 6B), with the exemption of Casein kinase I/II motif. Casein kinase II has been shown in a recent study by our group to be involved in microtubule-based protrusions in the context of the binary toxin CDT of *Clostridioides difficile* [49]. Since no pyknotic effect has hitherto been observed for CDT, we suggest that the upregulation is primarily due to the morphological changes during necrosis, although no microtubule-based protrusions have been shown for TcdB effects so far. To elucidate the impact of the RAS inhibition effect on the signaling cascade of the glucosyltransferase-independent effect of TcdB, especially at the early time points, other phosphoproteomic studies should be helpful.

Noticeably, the only protein that responded with a 2-fold significant change to both TcdB and TcdB_NXN_ after 3144 pretreatment was the apolipoprotein C-III (APOC3). APOC3 was found to be significantly upregulated in a time-dependent manner as a response to only TcdB or TcdB_NXN_ treatment (Figure 7A). Its expression was 2-fold significantly decreased when cells were pretreated with 3144 prior to TcdB or TcdB_NXN_ treatment (Figure 7B). APOC3 is a secreted protein that has most recently been shown to activate the NLRP3 inflammasome in monocytes, promoting tissue damage and broad inflammation response [23]. Moreover, NLRP3 inflammasome activation has been shown in macrophages treated with TcdB, but no activation was obvious when treated with TcdA [50]. Taken together, these findings point towards APOC3 as a potential drug target to reduce inflammation in myeloid-derived cells in vivo and link the glucosyltransferase-independent effect to a broader understanding of the CDI setting and the subsequent inflammatory diseases. Further investigations are necessary to investigate the role of APOC3 in vivo that points towards a deeper mechanistic understanding of the effect of TcdB.

Taken together, in this study, we conducted a deep phosphoproteomic analysis on the impact of TcdB and TcdB_NXN_ on HEp-2 cells. Through comparison with the phosphoproteomic effects of TcdA on HEp-2 cells for the first time and deep phosphoproteome kinetics for TcdB- and TcdB_NXN_-mediated effects, we were able to elucidate RAS as a central upstream regulator of the glucosyltransferase-independent effect. We showed elevated RAS activation shortly after TcdB treatment and noticed that RAS inhibition led to a strong and significant reduction in the glucosyltransferase-independent effect of TcdB. Further phosphoproteomic and proteomic investigation of the RAS-mediated effect revealed that APOC3 expression was induced by TcdB and TcdB_NXN_ and that RAS inhibition prohibited the expression of APOC3 upon TcdB and TcdB_NXN_ treatment. These findings contribute to a broad understanding of the TcdB-mediated inflammatory effects and will promote our knowledge of the full picture of CDI-mediated pathogenesis.

## 4. Materials and Methods

### 4.1. Cell Culture

Epithelial cell line HEp-2 (derived from HeLa cells) was maintained in a 75 cm^2^ flask in a humidified atmosphere at 37 °C and 5% CO_2_. Cells were cultured in Minimal Essential Medium (MEM) supplemented with 10% fetal bovine serum (FBS) and 100 units/mL Penicillin and 100 units/mL Streptomycin. Cells were split to maintain vitality depending on confluency. Two days before toxin treatment, 7.5 × 10^5^ cells were seeded in 10 cm dishes to achieve a 75% confluency on the day of toxin treatment. VPI10463 TcdB, TcdB_NXN_, TcdA and TcdA_NXN_ were generated using an *B. megaterium* expression system, as previously described [51]. The group of Harald Genth kindly provided TcsL. Cells were treated with 2 nM TcdB, TcdB_NXN_ or 20 nM TcdA, TcdA_NXN_ throughout all experiments. For preincubation with TcdA, TcdB, TcsL or the pan-RAS inhibitor 3144, 2.5 × 10^5^ cells were seeded into 6-well plates in 2 mL medium two days prior toxin treatment. Furthermore, 17 h before toxin treatment, cells were treated with 20 nM TcdA, 0.1 pM TcsL or 1 µM 3144 in a final medium volume of 2.5 mL without medium exchange, respectively.

Changes in the morphology were documented by phase-contrast microscopy after 8 h. For controls, only medium was exchanged, except for 3144 controls, where DMSO was added to the medium. After documentation, cells were washed twice with ice-cold PBS and lysed by scraping them into 600 µL lysis buffer (8 M Urea, 50 mM ammonium bicarbonate (pH 8.0), 1 mM sodium ortho-vanadate, complete EDTA-free protease inhibitor cocktail (Roche, Basel, Switzerland) and phosphoSTOP phosphatase inhibitor cocktail (Roche, Basel, Switzerland). Lysates were sonicated on ice two times for 5 s at 30% energy. Cell debris was removed by centrifugation for 15 min at 16,100× *g* at 4 °C. The supernatant was collected and used for further preparation.

### 4.2. Protein Digestion

BCA assay (Thermo Scientific, Waltham, MA, USA) was used to estimate protein concentrations. Proteins were reduced with 5 mM DTT at 37 °C for 1 h and afterwards alkylated with 10 mM iodoacetamide at RT for 30 min. DTT was added to a final concentration of 5 mM to quench alkylation. Lysates were diluted 1:5 with 50 mM ammonium bicarbonate (ABC) buffer to lower the concentration of urea below 2 M for digestion. Furthermore, 1 mg of protein per condition was digested at 37 °C for 4 h using 1:100 *w*/*w* endoproteinase Lys-C (Wako, Osaka, Japan), followed by overnight digestion with 1/100 *w*/*w* trypsin (Promega, Madison, MI, USA). Digestion was stopped by adding Trifluoroacetic acid (TFA) to a final concentration of 1%. Peptide solutions were then desalted with Sep Pak C18 1 cc cartridges (Waters).

### 4.3. Phosphopeptide Enrichment and TMT Labeling

Before phosphopeptide enrichment, 25 µg of peptide of each sample was set aside for the proteome measurement. A High-Select™ TiO2 Phosphopeptide Enrichment Kit (Thermo Scientific, Waltham, MA, USA) was used to enrich phoshphopeptides followed by a second enrichment step for the flowthrough fraction of the TiO_2_ kit using the High-Select™ Fe-NTA Phosphopeptide Enrichment Kit (Thermo Scientific, Waltham, MA, USA) according to manufacturer’s instructions. Afterwards, enriched phosphopeptides were desalted by C18 spin tips. BCA assay was used before labeling to estimate the peptide concentration of each sample. Equal amounts of enriched phosphopeptides and peptides for the proteomic analysis were labeled and combined as previously described [52]. After labeling, peptides were fractionated into eight fractions utilizing a high-pH fractionation kit (Thermo Scientific, Waltham, MA, USA). For the analysis of the effect of the pan-RAS inhibitor on the phosphoproteome, no fractionation was performed. Finally, peptides were vacuum dried and stored at −80 °C until LC–MS measurement.

### 4.4. LC–MS Analysis

Samples were dissolved in 0.1% TFA/2% acetonitrile (ACN) and analyzed in an Orbitrap Fusion Lumos mass spectrometer (Thermo Scientific, Waltham, MA, USA) equipped with a nanoelectrospray source and connected to an Ultimate 3000 RSLC nanoflow system (Thermo Scientific, Waltham, MA, USA). Peptides were loaded on an Acclaim PepMap C18 trap (Thermo Scientific, Waltham, MA, USA) and separated by a 50 cm µPAC™ (PharmaFluidics, Gent, Belgium) analytical column at 35 °C column temperature, utilizing 0.1% formic acid solvent as solvent A and 100% ACN with 0.1% formic acid as solvent B. We used a 120 min gradient at a flow rate of 500 nL/min, increasing it from 3.4% B to 21% B in 65 min, to 42% B in 32 min and to 75.6% B in 2 min; maintaining it for 3 min; then decreasing it to 3.6% B in 2 min; and maintaining it for 16 min. The spray voltage was set to 2 kV. A data-dependent acquisition method was used with a cycle time of 3 s and top N setting. Dynamic exclusion was set to 60 s, an AGC target of 4 × 10^5^, a maximum injection time of 50 msec and an Orbitrap resolution of 120,000 for MS1 scan. For all runs, an MS2 method was used with HCD fragmentation at 38%, first mass at 100 *m*/*z*, an MS^2^ maximum injection time of 110 msec, an MS^2^ isolation width of 0.8 *m*/*z* and an Orbitrap resolution of 60,000.

### 4.5. Flow Cytometry Analysis

The annexin-V-APC/propidium iodide kit (MabTag, Friesoythe, Germany) was used to assess the percentage of necrotic and apoptotic cells after toxin treatment by flow cytometry. All reagents were used according to the manufacturer’s instructions. In brief, cells were detached from the well bottom utilizing a trypsin/EDTA solution and spun down for 5 min at 300 g; 100 µL of annexin binding buffer and the annexin-APC solution were added and incubated at RT for 30 min. Directly before measurement, propidium iodide was added utilizing the autolabel function of the MACSQuant^®^ Analyzer 10 (Miltenyi Biotec, Bergisch Gladbach, Germany).

### 4.6. DAPI Staining

For DAPI staining, attached cells were washed once with 500 µL PBS, and incubated for 10 min at 37 °C and 5% CO_2_ with 500 µL of a 300 nM DAPI/PBS solution. Afterwards, cells were washed twice with 500 µL PBS and analyzed via fluorescence and phase contrast microscopy.

### 4.7. RAS Activation Assay

To measure the RAS activation status, a pull down of the GTP-bound form of the three different isoforms of RAS (H, K and N) was performed by utilizing a GST fusion protein of the RAS-binding domain of RAF1 as a specific probe for activated RAS, as previously described [53].

### 4.8. Western Blotting

Cells were lysed with 500 µL RIPA-Lysis buffer supplemented with phosphatase and proteinase inhibitors [53], and homogenized utilizing a 0.45 × 25 mm needle on a syringe. Debris was removed by centrifugation at 16,100× *g* for 15 min at 4 °C. Subsequently, 50 µg of protein was applied for Western blotting. For total-MAPK1/3 detection, p44/42 MAPK (Erk1/2) (L34F12) (#4696), and for phospho-MAPK1/3, rabbit anti-Phospho-p44/42 MAPK (Erk1/2) (Thr202/Tyr204) (D13.14.4E) XP (#4370), from Cell Signaling were used. RAS was detected using pan-RAS (Ab-3) (#OP40) from Merck (Darmstadt, Germany). Protein detection was performed using the Odyssey Sa Infrared Imaging System (LI-COR Bioscience, Lincoln, NE, USA) or enhanced chemiluminescence (ECL, Thermo Scientific, Waltham, MA, USA). ImageStudio (LI-COR) software was employed for quantitative analysis.

### 4.9. Data Processing

Raw data were processed with MaxQuant software (version 1.6.3.3) [54] using the Andromeda search engine [55]. Spectra were searched against human entries of the Swiss-Prot reviewed UniprotKB database (version 01/2021) [56] and controlled by FDR < 0.01 at the peptide and protein level. Carbamidomethylation of cysteine was set as fixed modification, and as variable modifications, oxidation of methionine, N-terminal acetylation, deamidation of glutamine and asparagine were set. Phosphorylation (PO4) was set at serine, threonine and tyrosine residues as variable modifications. The maximum of missed cleavage was set to 2. Only phosphosites and proteins were used for quantification, which were measured in three replicates. Measured phosphosites with a localization probability below 75% were excluded from further processing. Both the proteome and phosphoproteome were normalized by subtracting the mean intensity of each sample and the mean intensity of each TMT-batch, respectively. Only phosphosites that could be normalized using the corresponding protein were later included in quantitative analysis. Except for the analysis of the pan-RAS inhibitor 3144 phosphoproteome, all phosphosites that had a localization probability above 75% were included in the analysis. Data evaluation, analysis and visualization were performed using Perseus (version 1.6.2.3.) [57]. The upstream analysis was generated through ingenuity pathway analysis (QIAGEN, Hilden, Germany). All significantly regulated sites were included. R [58], in particular, the R packages complex heatmap [59] and ggplot2 [60], was used for data analysis and visualization. For heatmap generation, Benjamini Hochberg FDR-based ANOVA tested protein and phosphosite intensities were used. For gene ontology and network analysis, the STRING db [61] database, and for visualization, Cytoscape [62], were used. The mass spectrometry proteomics data have been deposited in the ProteomeXchange Consortium via PRIDE [63] partner repository with the dataset identifier PXD031726.

## Figures and Tables

**Figure 1 ijms-23-04258-f001:**
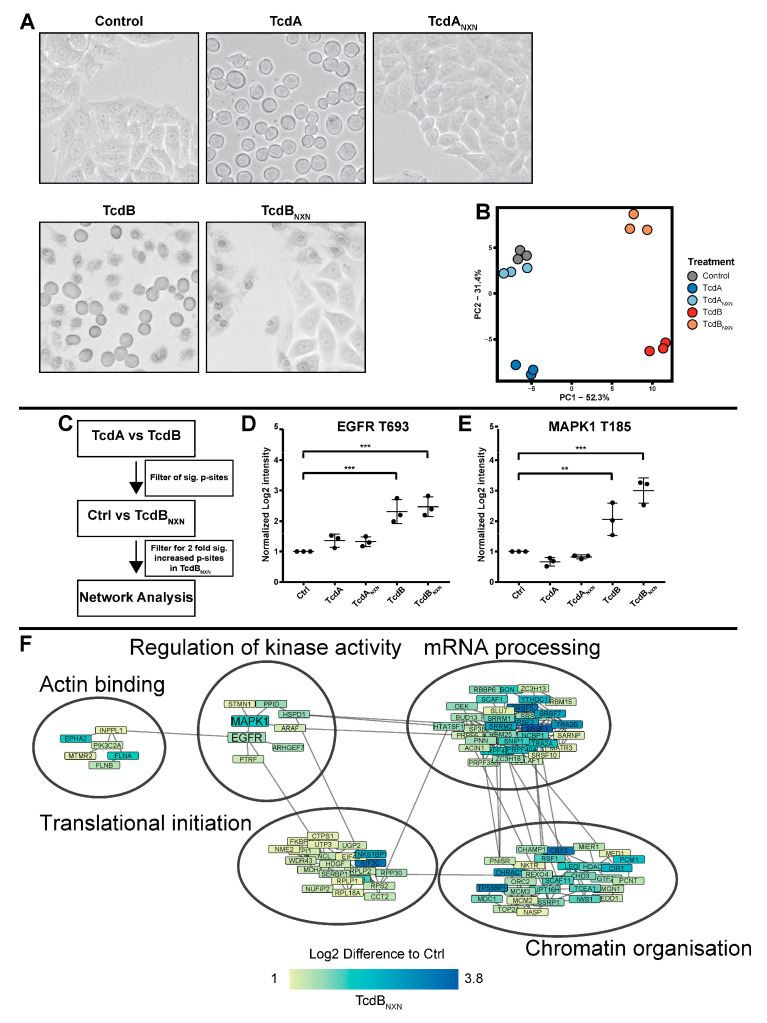
Phosphoproteome analysis of HEp-2 cells treated with TcdA, TcdA_NXN_, TcdB or TcdB_NXN_. Cells were treated for 8 h with 20 nM of TcdA or TcdA_NXN_ or 2 nM of TcdB or TcdB_NXN_. (**A**) Morphological changes in differently treated HEp-2 cells. (**B**) Principal component analysis (PCA) of significantly (FDR-based ANOVA *p* < 0.05) altered phosphosites (PC1: principal component 1; PC2: principal component 2). (**C**) Schematic overview of iterations made to identify the phosphosites that were responsive to the glucosyltransferase-independent effect and contribute to the underlying network. Changes in phosphorylation for (**D**) EGFR at site T693 and (**E**) MAPK1 at site T185. (**F**) Network of glucosyltransferase-independent effects. (*p*-value: ** *p* > 0.01; *** *p* > 0.001; no * marking = no significance compared to control).

**Figure 2 ijms-23-04258-f002:**
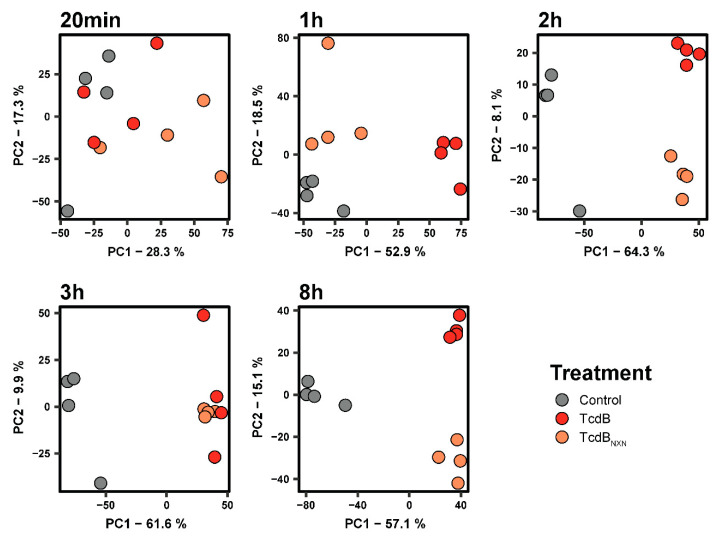
Principal component analysis for different time points of the phosphoproteome of untreated, TcdB- and TcdB_NXN_-treated HEp-2 cells.

**Figure 3 ijms-23-04258-f003:**
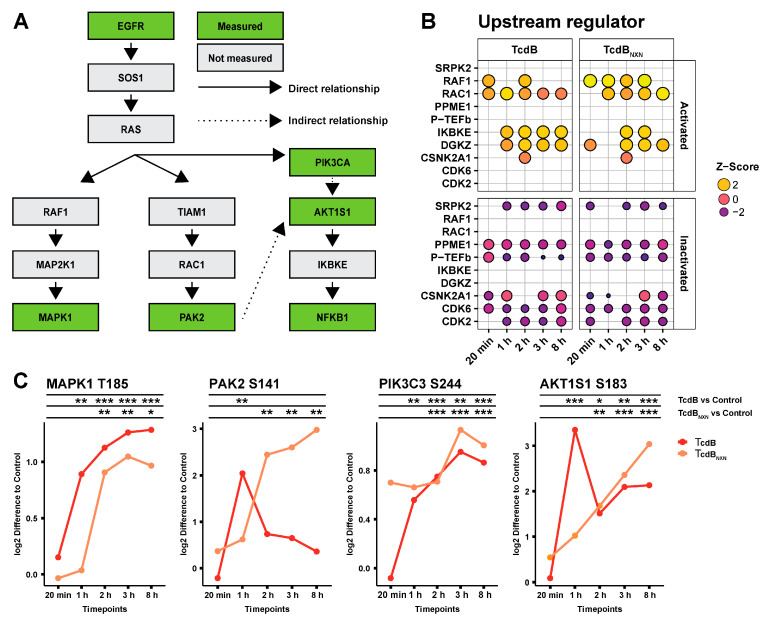
Overview of the signaling cascade after TcdB and TcdB_NXN_ treatment. (**A**) Schematic overview of the possible signaling cascade for the glucosyltransferase-independent effect. (**B**) Comparison of the top 10 upstream regulator of TcdB and TcdB_NXN_ after different treatment time points. (**C**) Selected phosphosites from the schematic overview shown in (**A**) comparing the effect of TcdB and TcdB_NXN_ after different time points. (*p*-value: * *p* > 0.05; ** *p* > 0.01; *** *p* > 0.001; no * marking = no significance compared to control).

**Figure 4 ijms-23-04258-f004:**
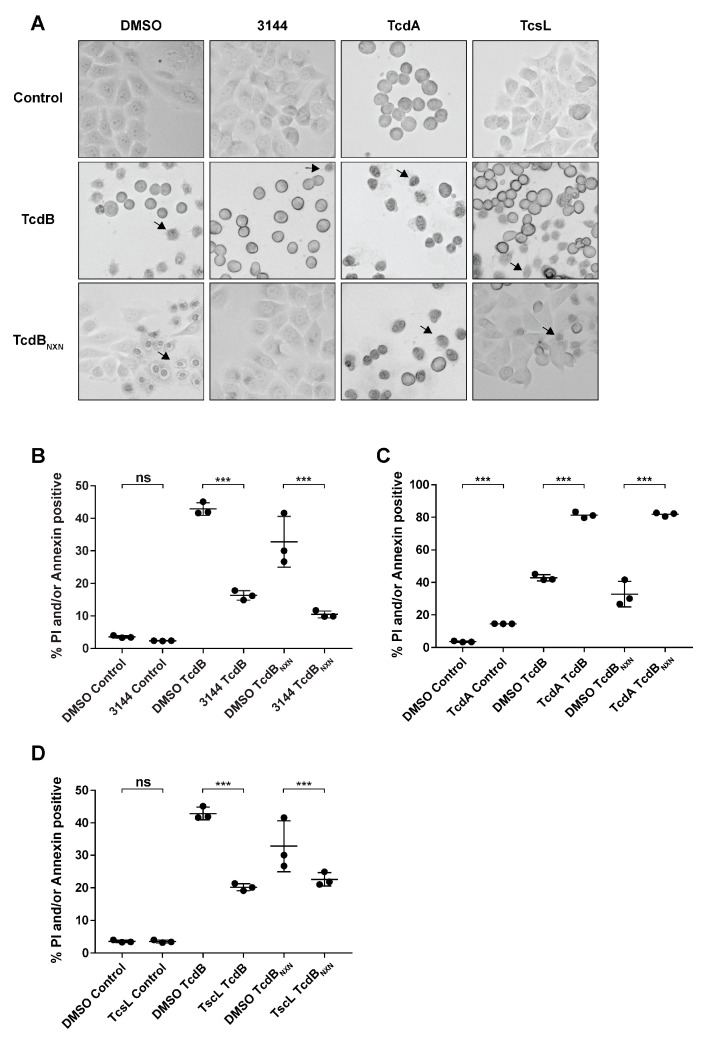
RAS inhibition reduces the necrotic phenotype after TcdB_NXN_ and TcdB treatment. (**A**) Changes in morphology of HEp-2 cells after 8 h of treatment with 2 nM TcdB or TcdB_NXN_ and 17 h pretreatment with 20 nM TcdA, 0.1 pM TcsL and 1 µM pan-RAS inhibitor 3144, respectively. Arrows indicate pyknotic cells. (**B**–**D**) Percentage of annexin- and/or propidium iodide-positive cells detected via flow cytometry after 8 h incubation time with 2 nM TcdB or TcdB_NXN_ and 17 h pretreatment with (**B**) 1 µM pan-RAS inhibitor 3144, (**C**) 20 nM TcdA, (**D**) 0.1 pM TcsL and DMSO control, respectively. n = 3. (*p*-value: *** *p* > 0.001, ns = not significant).

**Figure 5 ijms-23-04258-f005:**
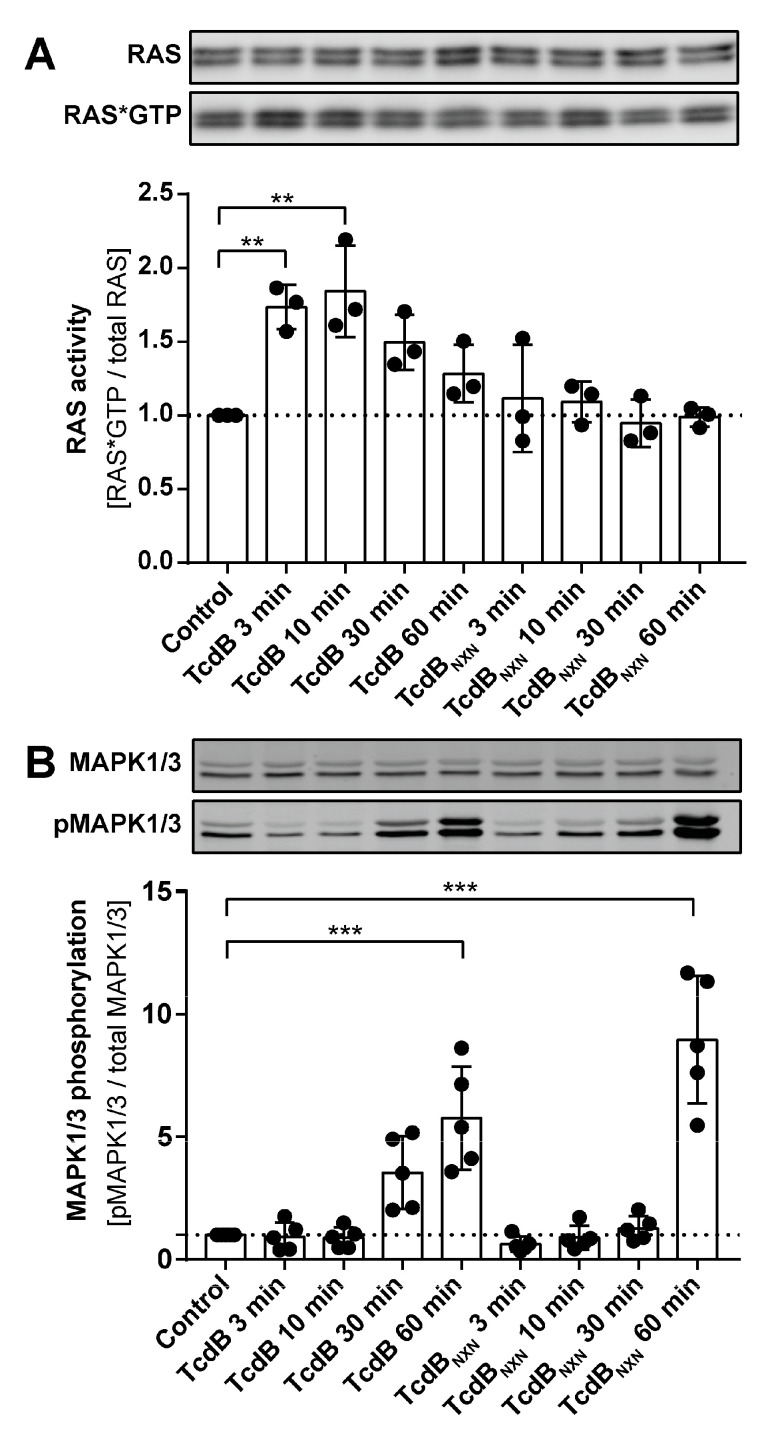
RAS activation assay. (**A**) The upper panel shows a representative Western blot of total RAS and RAS*GTP, and the lower panel shows the densitometry analysis of RAS*GTP pull-down experiment after treatment with TcdB or TcdB_NXN_ for different time points (n = 3). (**B**) A representative Western blot and the densitometric analysis of MAPK1/3 phosphorylation (n = 5). (*p*-value: ** *p* > 0.01; *** *p* > 0.001; no * marking = no significance compared to control).

**Figure 6 ijms-23-04258-f006:**
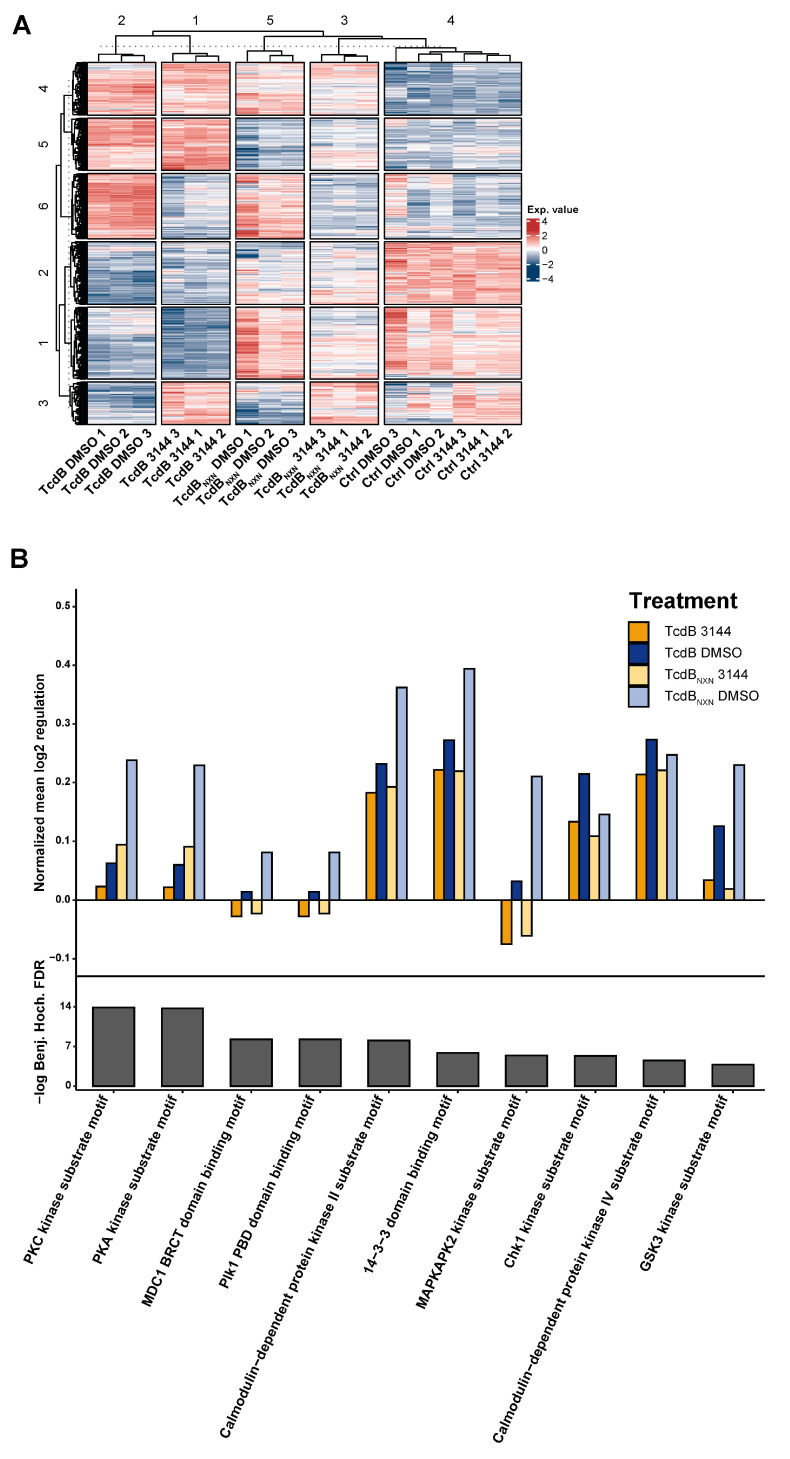
Phosphoproteome analysis of HEp-2 cells after treatment with TcdB or TcdB_NXN_ with pan-RAS inhibitor 3144 or with DMSO. (**A**) Heatmap of significantly changed phosphosites (Benjamini Hochberg FDR-based ANOVA, *p* < 0.05) after 8 h treatment with TcdB or TcdB_NXN_ and preincubation with 1 µM 3144 or DMSO for 17 h. (**B**) Mean phosphorylation of in Fisher’s exact test enriched “co-phosphorylated” motifs after 8 h TcdB or TcdB_NXN_ treatment with and without pan-RAS inhibitor 3144. ((**A**) Exp. Value: expression value).

**Figure 7 ijms-23-04258-f007:**
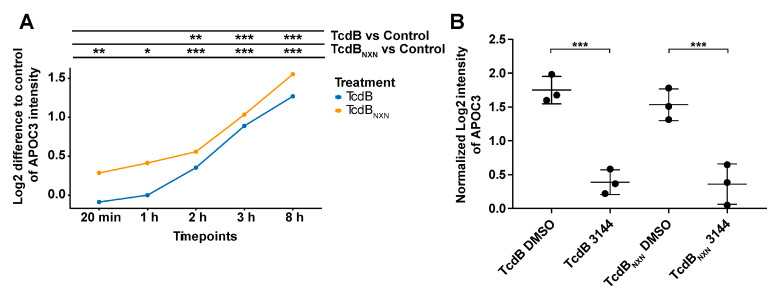
APOC3 protein expression after TcdB or TcdB_NXN_ treatment. (**A**) Expression difference of APOC3 to control over time after treatment with 2 nM TcdB or TcdB_NXN_. **(B**) APOC3-normalized expression after 8 h of treatment with 2 nM TcdB or TcdB_NXN_ and 17 h prior to incubation with 1 µM pan-RAS inhibitor 3144 or DMSO. Presented data show MS-based expression analysis. (*p*-value: * *p* > 0.05; ** *p* > 0.01; *** *p* > 0.001; no * marking = no significance compared to control).

## Data Availability

The mass spectrometry proteomics data have been deposited in the ProteomeXchange Consortium via Pride partner repository with the dataset identifier PXD031726.

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
