# Peer review of "TcdB of Clostridioides difficile Mediates RAS-Dependent Necrosis in Epithelial Cells"

_ijms, 2022, doi:10.3390/ijms23084258_

Round 1

Reviewer 1 Report

The study investigates signaling mechanisms underlying epithelial cell death induced by Clostridial toxins. The authors performed large scale phospho-proteomic analysis and also identified the roles of Ras family of small GTPases in necrotic cell death.

The study is well-designed and manuscript is well-written. Presented results are robust and well-interpreted. I have two minor comments/suggestions.

  1. I am surprised that densitometric quantification of MAP kinase activation in toxin-treated cells did not statistically significant increase in MAPK signaling (phosphorylation). The presented blots show clear upregulation of MAPK phosphorylation. The authors may consider either showing more representative blot or re-evaluating their quantification analysis.
  2. Figure 7 should include original immunoblots showing APOC3 expression, not just quantification of the signal intensity. Also, the Materials part of the manuscript does not indicate which APOC3 antibody was used.

Author Response

"TcdB of Clostridioides difficile mediates RAS-dependent ne-crosis in epithelial cells": Detailed Response to the Editor and the Reviewers

We sincerely appreciate the valuable comments and suggestions from the reviewers. The thorough review helped immensely in the shaping of the manuscript. The suggestions and comments have been closely followed, and revisions have been made accordingly. The following are the questions extracted from the reviewers' comments and our responses. For the reviewers' convenience, the revisions have been marked using the “Track Changes” function in the manuscript as recommended by the journal.

Response to Reviewer 1

Comment 1: I am surprised that densitometric quantification of MAP kinase activation in toxin-treated cells did not statistically significant increase in MAPK signaling (phosphorylation). The presented blots show clear upregulation of MAPK phosphorylation. The authors may consider either showing more representative blot or re-evaluating their quantification analysis.

Response 1: Thank you for raising this important point. We agree with the reviewer that the western blot image showed a clear increase in phosphorylation. The main problem was that although the increase was very high, the activation varied greatly so that no statistical significance could be determined. We have now included two additional MAPK phosphorylation western blot results in our analysis, and the differences were highly significant. We still had additional western blots from RAS assays, where the corresponding pulldown was not evaluable. We changed the manuscript accordingly.

Comment 2: Figure 7 should include original immunoblots showing APOC3 expression, not just quantification of the signal intensity. Also, the Materials part of the manuscript does not indicate which APOC3 antibody was used.

Response 2: Thank you for allowing us to clarify. The APOC3 expression data were from mass spectrometry-based analysis and no immunoblotting was conducted. We changed the manuscript accordingly to make this clearer.

Reviewer 2 Report

see attachment.

Author Response

"TcdB of Clostridioides difficile mediates RAS-dependent ne-crosis in epithelial cells": Detailed Response to the Editor and the Reviewers

We sincerely appreciate the valuable comments and suggestions from the reviewers. The thorough review helped immensely in the shaping of the manuscript. The suggestions and comments have been closely followed, and revisions have been made accordingly. The following are the questions extracted from the reviewers' comments and our responses. For the reviewers' convenience, the revisions have been marked using the “Track Changes” function in the manuscript as recommended by the journal.

Response to Reviewer 2

Comment 3: The introduction states that "TcdB exhibits rapid necrotic death at high toxin concentrations in vitro called pyknosis" (p. 2 lines 48-49), but the authors use just 2 nM of TcdB for their studies. Is 2 nM considered a high toxin concentration? The authors should define what "high toxin concentrations" are in line 48.

Response 3: Thank you for giving us the opportunity for clarification. High toxin concentration meant a concentration above 0.1 nM. Above that level, pyknosis is induced. We have chosen 2 nM TcdB concentration because it provides reliantly a strong pyknotic effect during the analysis time that is needed for our phosphoproteomic study. We changed the manuscript accordingly to clarify this matter further.

Comment 4: Multiple experiments conclude pyknotic cell death is occurring through changes in cell morphology examined by light microscopy. However, the morphological changes characteristic of pyknosis are never defined.  How can the authors conclude from light microscopy alone that pyknosis is occuring?

Response 4: Thank you for pointing this out. Pyknosis is visually characterized by cell shrinkage, chromatin condensation and membrane blebbing. However, pyknosis morphology is very characteristic and easy to identify by light microscopy. We controlled the necrotic phenotype via DAPI staining as seen in supplementary figure 4. Unfortunately, due to technical problems the supplementary data was not uploaded with manuscript submission. In supplementary figure 4 DAPI staining confirmed additionally that cells exhibiting pyknotic phenotype were necrotic and, therefore DAPI positive. We added a definition to the introduction and a sentence on that matter to the results paragraph.

Comment 5: In Figures 4B-D, why did the authors use a method for detecting apoptosis (PI/Annexin V staining) to monitor the pyknotic/necrotic cell death induced by TcdB?   

Response 5: Thank you for asking this question. By using PI/Annexin staining, we could detect both apoptosis and necrosis to control possible apoptotic influences since late apoptotic cells could not be distinguished with only PI staining. Additionally, we used DAPI staining to monitor the pyknotic effect for DMSO and 3144 inhibitor (Supp. Fig. 4). No changes to the manuscript were made.

Comment 6: Why is there little to no toxicity observed in cells treated with TcdA alone (Figure 4C)?  As noted in the Introduction, TcdA causes an apoptotic effect that should be detected by the PI/Annexin V staining procedure of Figure 4C.

Response 6: Thank you for raising this question. Unfortunately, due to technical problems, the supplementary data were not uploaded. The apoptotic effect is clearly visible after treatment, as seen in supplementary figure 5. Additionally, TcdA causes apoptosis after 24-48h (Nicole M. Chumbler et al. 2016), consequently, the seen early-stage apoptotic effect is reasonable from our perspective. We changed the manuscript accordingly.

Comment 7: Most experiments in the paper use an 8 h toxin exposure, but Figure 5 limits toxin exposure to 1 h.  Why did the authors use such a relatively short time frame for the experiments of Figure 5. Due to the different time frames, is it possible to correlate the results of Figure 5 to other results in the paper?

Response 7: Thank you for pointing this out. We also performed experiments on longer time points and observed no further RAS activation. This is probably because RAS activation directs the signaling towards the pyknotic pathway in the first hour and activates thus downstream signals as calcium signaling and strong ROS production that then drives the pyknotic effect without further RAS activation. We changed the manuscript accordingly.

Comment 8: Inhibition of Ras reduces but does not eliminate the glucosyltransferase-independent toxicity of TcdB (Fig. 4).  Does this mean there is also a Ras-independent mechanism of cell death for the glucosyltransferase-independent toxicity of TcdB?  This should be considered by the authors.

Response 8: Thank you for bringing this point into question. We agree with the reviewer that a Ras-independent mechanism could exist. Although Ras-dependent mechanisms are difficult to study since complete abolishment of Ras signaling is deadly for cells, we titrated the RAS inhibitor to the point where reduction of the pyknotic effect was at a maximum and the induction of cell death was not yet present. Therefore, we suggest that a partial inhibition of Ras signaling already leads to a reduction in the pyknotic effect and thus, no complete elimination of the pyknotic effect can be achieved by this method. Further studies should evaluate whether a specific Ras isoform or a broad-based Ras effect is responsible for the pyknotic effect. If it is a specific Ras isoform, it might be feasible to perform a knockdown experiment leaving the cells alive and enable to analyze the glucosyltransferase independent effect further. Nevertheless, a Ras independent mechanism might be possible. We changed the manuscript accordingly. 

Comment 9: Why does TcdA sensitize cells to the glucosyltransferase-independent function of TcdB (Fig. 4C)? The authors do not appear to provide a potential explanation for this phenomenon.

Response 9: Thank you for pointing this out. A most reasonable explanation is that TcdB enhances apoptotic processes as seen in the supplementary figure where an apoptotic population below the necrotic one emerged, leading to a higher PI/Annexin positive rate. Changes to the manuscript has been made accordingly.

Comment 10: The activation of Ras within 3 minutes of exposure to wild-type TcdB seems extremely odd (Figure 5A). It is highly unlikely that the exogenous application of TcdB would, in just 3 minutes, result in its surface binding, endocytosis, translocation of the catalytic subunit to the cytosol, and sufficient cytosolic toxin activity to activate Ras.  Perhaps Ras activation is simply due to toxin binding at the cell surface?  Did the authors run any control conditions to ensure Ras activation required toxin translocation to the cytosol (ie, is it inhibited by endosomal alkalinization?). 

Response 10: Thank you for raising this point. We agree with the reviewer that receptor binding could possibly induce Ras activation but we did not control for translocation or receptor binding. This is an excellent point for future studies investigating the main cause of Ras upstream signaling and activation. Nevertheless, since TcdBNXN was found to be less potent than the wild type TcdB in causing the pyknotic effect and Ras activation for TcdB was only narrowly visible, Ras activation of TcdBNXN could be below the detection limit. If Ras activation was receptor-mediated, TcdB and TcdBNXN should have produced the same activation signal since the mutations of TcdBNXN are not in the known receptor binding region of TcdB but in the glucosyltransferase domain. Changes to the manuscript have been made accordingly.

Comment 11: In Figure 5, the authors detect MAPK1/3 activation after 60 min of exposure to TcdB(nxn) but do not detect Ras activation by TcdB(nxn) over the same time course.  Is the activation of MAPK1/3 by TcdB(nxn) ras-independent?  This seems to be the straightforward interpretation of the presented data, but it does not seem to be the conclusion of the authors.

Response 11: Thank you for pointing this out. We agree with the reviewer a possible activation of MAPK1/3 could be independent of RAS activation. Nevertheless, MAP kinase pathway dependence on RAS could clearly be seen to be RAS dependent since MAP kinase-dependent motifs were upregulated after TcdB and TcdBNXN treatment and inhibited after incubation with RAS-inhibitor prior to TcdB and TcdBNXN (Figure 6). We changed the manuscript accordingly.

Comment 12: The authors conclude that Ras is active in the glucosyltransferase-independent function of TcdB, but they could not detect Ras activation by the glucosyltransferase-deficient TcdB(nxn) variant of the toxin (Fig. 5A).

Response 12: Thank you for your comment. We agree with the reviewer that Ras activation by TcdBNXN was not visible. A possible explanation could be that since we found TcdBNXN to be less potent than the wild type TcdB in causing the pyknotic effect and the RAS activation for TcdB was only narrowly visible. RAS activation of TcdBNXN could be below the detection line or since the pyknotic effect of TcdBNXN was delayed compared to the wild type, we don’t have met the right time point. We changed the manuscript accordingly.

Comment 13: Is it possible that the pan-ras inhibitor 3144 has off-target effects that could muddle the conclusion of Ras dependency for the glucosyltransferase-independent function of TcdB?

Response 13: Thank you for raising this essential point. Since small GTPases have a very high homology, a possible off-target effect on the other GTPases might be possible. Therefore, we controlled the effect of the inhibitor with TcdA preincubation, that glucosylates other GTPases for example Rac1 far more efficient than Ras (Junemann et al. 2017). Furthermore, we used TcsL which main target is Ras as positive control (Genth et al. 2018). We also performed preincubation with 0.03 nM TcdB preincubation since TcdB cannot glucosylate Ras at all (Genth et al. 2018). Preincubation with 0.03 TcdB showed the same results as TcdA; therefore was not included in the manuscript. Moreover, TcdBF an isoform of TcdB that glucosylate Ras has been shown to exhibit no pyknotic effect (Wohlan et al. 2014). Thus, we are confident that Ras plays a role in the glucosyltransferase independent effect with the possibility of other Ras-independent pathways for pyknosis. We changed the manuscript accordingly. 

Comment 14: A TcdB concentration of 2 nM is listed for several experiments, but other experiments do not list a toxin concentration.  It is assumed that 2 nM is used throughout the paper.  If another TcdB concentration was used for any experiment, it should be noted in the text and figure legend.

Response 14: Thank you for pointing this out. You are right no other concentration were used. We changed the material and methods part accordingly to clarify this.

Comment 15: p. 9 line 199:  "Figure 5A" is actually referring to Figure 5B.

Response 15: Thank you for pointing this out. We changed the manuscript accordingly.

 Comment 16: p. 9 lines 200-202: the result described as "data not shown" appears to be shown in Figure 5A.

Response 16: Thank you for pointing this out. We changed the manuscript accordingly.

Comment 17: The legend for Figure 7 has reversed panels A and B.  The description of panel A is actually referring to panel B, and the description of panel B is actually referring to panel A.

Response 17: Thank you for pointing this out. We changed the manuscript accordingly.

Comment 18: The role of APOC3 is explained in the Discussion, but there should be at least a one-sentence description of the protein when it is first mentioned in the Results (p. 12 line 251).

Response 18: Thank you for pointing this out. We added a sentence in the described section.

Literaturverzeichnis

Genth, Harald; Junemann, Johannes; Lämmerhirt, Chantal M.; Lücke, Arlen-Celina; Schelle, Ilona; Just, Ingo et al. (2018): Difference in Mono-O-Glucosylation of Ras Subtype GTPases Between Toxin A and Toxin B From Clostridioides difficile Strain 10463 and Lethal Toxin From Clostridium sordellii Strain 6018. In: Frontiers in microbiology 9, S. 3078. DOI: 10.3389/fmicb.2018.03078.

Junemann, Johannes; Lämmerhirt, Chantal M.; Polten, Felix; Just, Ingo; Gerhard, Ralf; Genth, Harald; Pich, Andreas (2017): Quantification of small GTPase glucosylation by clostridial glucosylating toxins using multiplexed MRM analysis. In: Proteomics 17 (9). DOI: 10.1002/pmic.201700016.

Nicole M. Chumbler; Melissa A. Farrow; Lynne A. Lapierre; Jeffrey L. Franklin; D. Borden Lacy (2016): Clostridium difficile Toxins TcdA and TcdB Cause Colonic Tissue Damage by Distinct Mechanisms. In: Infection and Immunity 84 (10), S. 2871–2877. DOI: 10.1128/IAI.00583-16.

Wohlan, Katharina; Goy, Sebastian; Olling, Alexandra; Srivaratharajan, Sangar; Tatge, Helma; Genth, Harald; Gerhard, Ralf (2014): Pyknotic cell death induced by Clostridium difficile TcdB: chromatin condensation and nuclear blister are induced independently of the glucosyltransferase activity. In: Cellular microbiology 16 (11), S. 1678–1692. DOI: 10.1111/cmi.12317.

Round 2

Reviewer 2 Report

My comments have been appropriately addressed.